# Generation of the transgene-free canker-resistant *Citrus sinensis* using Cas12a/crRNA ribonucleoprotein in the T0 generation

Hang Su [1], Yuanchun Wang [1], Jin Xu [1], Ahmad A. Omar [2,3], Jude W. Grosser[2], Milica Calovic[1], Liyang Zhang[4], Yu Feng[1], Christopher A. Vakulskas [4] & Nian Wang [1] ✉

Citrus canker caused by *Xanthomonas citri* subsp. citri (*Xcc*) is a destructive citrus disease worldwide. Generating disease-resistant cultivars is the most effective, environmentally friendly and economic approach for disease control. However, citrus traditional breeding is lengthy and laborious. Here, we develop transgene-free canker-resistant *Citrus sinensis* lines in the T0 generation within 10 months through transformation of embryogenic protoplasts with Cas12a/crRNA ribonucleoprotein to edit the canker susceptibility gene *CsLOB1*. Among the 39 regenerated lines, 38 are biallelic/homozygous mutants, demonstrating a 97.4% biallelic/homozygous mutation rate. No off-target mutations are detected in the edited lines. Canker resistance of the *cslob1*-edited lines results from both abolishing canker symptoms and inhibiting *Xcc* growth. The transgene-free canker-resistant *C. sinensis* lines have received regulatory approval by USDA APHIS and are exempted from EPA regulation. This study provides a sustainable and efficient citrus canker control solution and presents an efficient transgene-free genome-editing strategy for citrus and other crops.

Citrus canker caused by *Xanthomonas citri* subsp. citri (*Xcc*) causes severe yield, quality and economic loss to citrus production worldwide and is endemic in most citrus-producing countries, such as U.S., Brazil, and China[1]. *Xcc* encodes a pathogenicity factor PthA4[2,3], a transcription activator-like effector (TALE) secreted by the type III secretion system. PthA4 enters the nucleus of plant cells to activate the canker susceptibility gene *LOB1* by binding to the effector binding elements in its promoter region, which subsequently induces expression of downstream genes and causes typical canker symptoms including hypertrophy and hyperplasia[3]. All commercial citrus cultivars are susceptible to citrus canker[4,5]. Citrus canker control relies primarily on treatment with copper-based antimicrobials[6], which cause environmental pollution[7–9]. Furthermore, copper-resistant *Xcc* strains have been reported in citrus producing regions where copper was frequently

used to control citrus canker[10,11]. Traditional breeding to generate canker-resistant citrus varieties has been hindered by heterozygosity, long juvenile period, self- and cross-incompatibility. The average breeding duration from the cross to the release of a cultivar for traditional citrus breeding requires approximately 20 years[12]. Transgenic expression of antimicrobial peptides[13–15], toxin[16], resistance genes[17,18], and immune-related genes[19–25] enabled increased canker resistance. Recently, CRISPR/Cas-mediated genome editing of the promoter or coding region of *LOB1* has conferred citrus resistance to *Xcc*[26–32]. However, the citrus plants generated by transgenic overexpression and CRISPR genome editing approaches were all transgenic. Transgenic crops face many challenges to be used in production owing to regulations and public perception concerns[33,34]. Consequently, none of the citrus plants generated by biotechnological approaches have been

[1]Citrus Research and Education Center, Department of Microbiology and Cell Science, Institute of Food and Agricultural Sciences, University of Florida, Lake Alfred, FL, USA. [2]Citrus Research and Education Center, Institute of Food and Agricultural Sciences, University of Florida, Lake Alfred, FL, USA. [3]Biochemistry Department, Faculty of Agriculture, Zagazig University, Zagazig, Egypt. [4]Integrated DNA Technologies, Inc, Coralville, IA, USA. ✉e-mail: nianwang@ufl.edu

registered and commercialized despite the tremendous effort and superior disease resistance.

Cas9 and Cas12a DNA, RNA or ribonucleoprotein complex (RNP) were successfully used to generate transgene-free crops in the T0 generation[35,36], which significantly shortens the time for plant genetic improvement by avoiding the lengthy process needed to remove transgenes. Specifically, the Cas/gRNA RNP method does not involve DNA fragments and has been used to generate transgene-free tobacco, Arabidopsis, lettuce, rice[36], Petunia[37], grapevine, apple[38], maize[39], wheat[40], and potato[41]. The RNP method is also known to reduce off-target mutations. For instance, off-target mutations were not detected for genome-edited maize[39] and wheat[40] that were generated by the RNP method. However, RNP-mediated genome editing efficacy is low[42].

In this work, we generate transgene-free canker-resistant *C. sinensis* cv. Hamlin (a widely planted citrus cultivar) lines using LbCas12a/crRNA RNP within 10 months (Fig. 1). Off-target mutations are not detected in the edited lines. Importantly, among the 39 regenerated lines, 38 lines are biallelic/homozygous mutants. The high efficacy and short time needed for Cas12a/crRNA RNP-mediated citrus genome editing will impact how citrus and other tree crops are genetically improved as well as their genetic studies in the future.

## Results
### Genome editing efficacy of Cas12a/crRNA RNP
To evaluate the transgene-free citrus genome editing efficacy of the RNP method, we first used the *CsPDS* gene (*orange1.1t02361*) as the target owing to its obvious albino phenotype which expedited the identification of biallelic/homozygous mutations[43]. Both Cas9 and Cas12a were successfully used in genome editing with the RNP method[36,40,44]. Here, we selected Cas12a because it generates longer deletions than Cas9[26,45]. We assessed both ErCas12a and LbCas12a-Ultra (hereafter LbCas12aU) in the RNP-mediated genome editing of embryogenic citrus protoplasts. Both ErCas12a and LbCas12a were

reported to have high genome editing efficacy[46,47]. LbCas12aU is a variant of LbCas12a and has increased genome editing efficacy than LbCas12a. We first evaluated their efficiency via in vitro digestion of a 555 bp DNA fragment from the first exon of the *CsPDS* gene (Fig. 2A). Both ErCas12a and LbCas12aU were able to digest the DNA fragments efficiently and generated two DNA fragments with the expected sizes of 320 bp and 240 bp (Fig. 2B, C). LbCas12aU showed a slightly higher efficiency than ErCas12a in in vitro digestion (Fig. 2B, C).

Next, LbCas12aU/crRNA and ErCas12a/crRNA RNPs were used to transform embryogenic *C. sinensis* cv. Hamlin protoplasts using the PEG method[48], which were used for plant regeneration without herbicide or antibiotics selection (Fig. 1). PCR amplification and Sanger sequencing analysis of RNP transformed protoplasts at 3 days post transformation (DPT) for *PDS* editing showed 14.3% and 16.7% mutation rate for LbCas12aU/crRNA and ErCas12a/crRNA, respectively. Six months after transformation with ErCas12a, 58 embryos showing an albino phenotype were selected for further analysis (Fig. 3A). Sanger sequencing analysis of the *CsPDS* gene indicated that the 58 albino mutants consisted of 56 homozygous mutants, 1 biallelic mutant, and 1 chimeric mutant. A randomly selected green embryo contained no mutations at the target site (Fig. 3). Among the homozygous mutants, 30 contained 7 bp deletion, whereas 26 contained 13 bp deletion. The biallelic mutant T0$_{Er}$−22 contained both 7 bp and 13 bp deletions, and the chimeric mutant T0$_{Er}$−20 contained 7 bp, 8 bp and 13 bp deletions (Fig. 3B). For LbCas12aU, 15 embryos showing an albino phenotype were selected for Sanger sequencing analyses (Fig. 3A). The sequencing result demonstrated that all the embryos generated from LbCas12aU/crRNA RNP transformation contained the same mutation type (9 bp deletion) at the target site (Fig. 3B). Thus, we concluded that both LbCas12aU/crRNA and ErCas12a/crRNA RNP transformation of embryogenic citrus protoplasts were able to efficiently generate biallelic/homozygous *CsPDS* mutations for *C. sinensis*.

In previous studies, RNP transformation showed low off-target mutations in plants[39,40]. One off-target sequence was identified for the

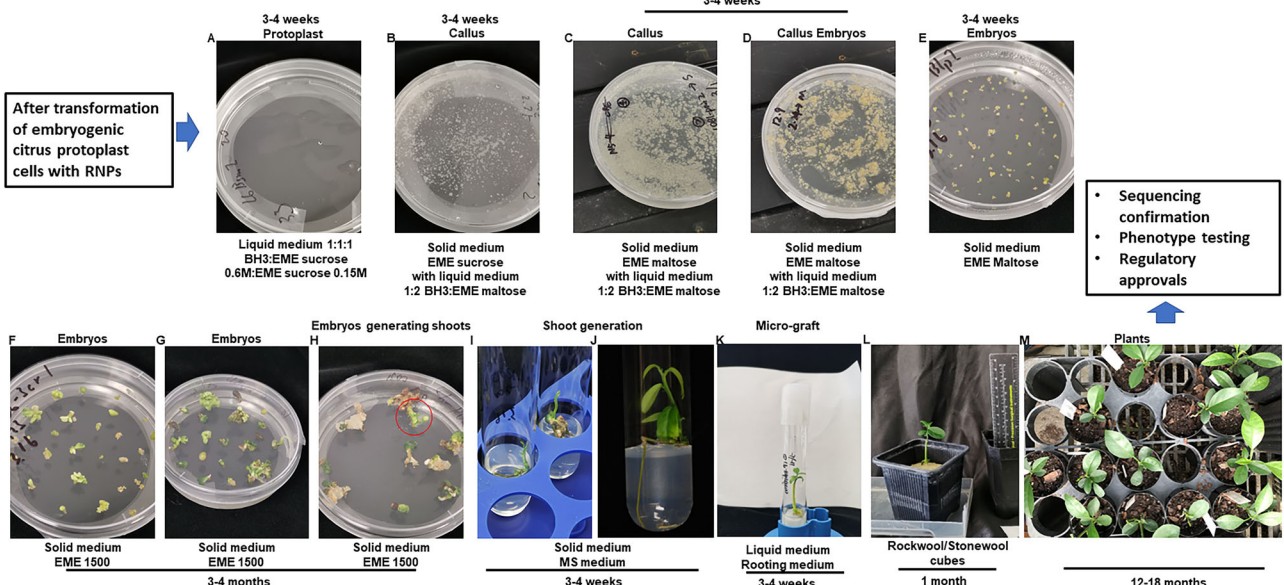

**Fig. 1 | Regeneration of genome-edited citrus protoplast cells. A** Edited citrus protoplast cells were kept in liquid medium (1:1:1 (v:v:v) mixture of BH3 and EME sucrose 0.6 M and EME sucrose 0.15 M) for 3–4 weeks at 28 °C in dark without shaking. **B** Citrus cells were transferred to EME sucrose medium added with 1:2 mixture of BH3 and EME maltose 0.15 M and kept at 28 °C for 3–4 weeks in dark. **C**, **D** Calli were transferred to EME maltose solid medium added with 1:2 mixture of BH3 and EME maltose 0.15 M and kept at 28 °C in dark for 3-4 weeks to generate embryos. **E** Embryos were transferred to EME maltose solid medium and kept at

room temperature under light for 3–4 weeks. **F**–**H** Embryos were transferred to solid EME1500 medium and kept at room temperature under light for 3–4 months to generate shoots. **I**, **J** Small plantlets were transferred to MS medium and kept at room temperature for 3–4 weeks. **K**–**M** The regenerated shoots were micro-grafted onto Carrizo citrange rootstock in liquid rooting media and kept in tissue culture room at 25 °C under light for 3–4 weeks (**K**), grown in stonewool cubes in growth chamber at 25 °C under light for 1 month (**L**), then planted in soil (**M**).

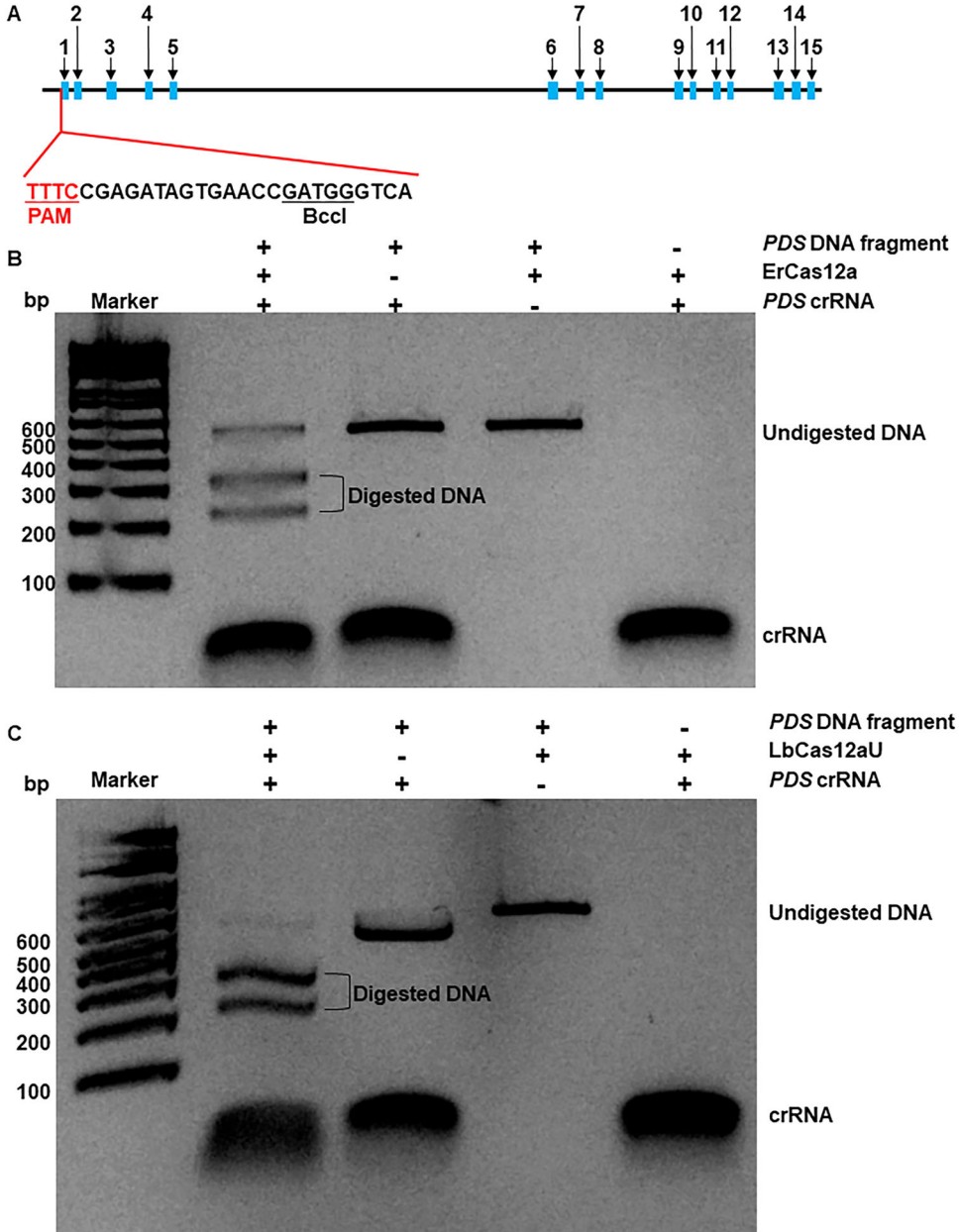

**Fig. 2 | Evaluate the crRNA guided endonuclease activity of Cas12a in vitro.**
**A** Schematic representation of the *CsPDS* gene (orange1.1t02361) and crRNA. Blocks in blue indicate exons. Line fragments indicate introns. TTTC in red: PAM (proto-spacer adjacent motif). BccI: restriction enzyme. **B, C** In vitro digestion of DNA fragments using ErCas12a (**B**) and LbCAs12aU (**C**). A DNA fragment (555 bp) of the *CsPDS* gene containing the crRNA target site as depicted in (**A**) was digested with ErCas12a (**B**) or LbCas12aU (**C**). The experiments in B and C were repeated at least two times with comparable results. After 30 min, DNA electrophoresis was run using 2% agarose gel. Source data are provided as a Source data file.

crRNA targeting *CsPDS* using the CRISPR P 2.0 system[49]. However, Sanger sequencing analyses of the 58 and 15 embryos generated by transformation of embryogenic protoplasts with ErCas12a/crRNA RNP and LbCas12aU/crRNA RNP, respectively, did not identify any muta-tions in the off-target homozygous sequence (Supplementary Fig. 1).

**Transgene-free genome editing of the canker susceptibility gene *CsLOB1***
We first tested the mutation rate of the *LOB1* gene for LbCas12aU/crRNA and ErCas12a/crRNA RNPs. PCR amplification and Sanger sequencing analysis of RNP transformed protoplasts at 3 DPT for *LOB1* editing showed 48.1% and 34.8% mutation rate for LbCas12aU/crRNA and ErCas12a/crRNA, respectively. The mutation rate seems to associate with the quality and status of the embryogenic protoplasts

because deep sequencing analysis of LbCas12aU/crRNA RNP trans-formed protoplasts from a different batch at 3 DPT demonstrated 71% mutation rate. In addition, LbCas12aU demonstrated superior activity in in vitro digestion of target sequence (Fig. 2). Thus, we used LbCas12aU/crRNA RNP in downstream studies to generate transgene-free canker-resistant *C. sinensis* cv. Hamlin by editing coding region of the canker susceptibility gene *CsLOB1*. We used one crRNA to target the 2nd exon of the *CsLOB1* gene (Fig. 4A) in the RNP complex. The crRNA was carefully designed to reduce off-target homo-logous sites.

In total 42 lines were regenerated (Fig. 1) and 39 lines survived in the greenhouse after micro-grafting on Carrizo. Surprisingly, PCR amplification and Sanger sequencing analyses of the 39 regener-ated lines demonstrated 38 lines were homozygous (8 lines)/biallelic

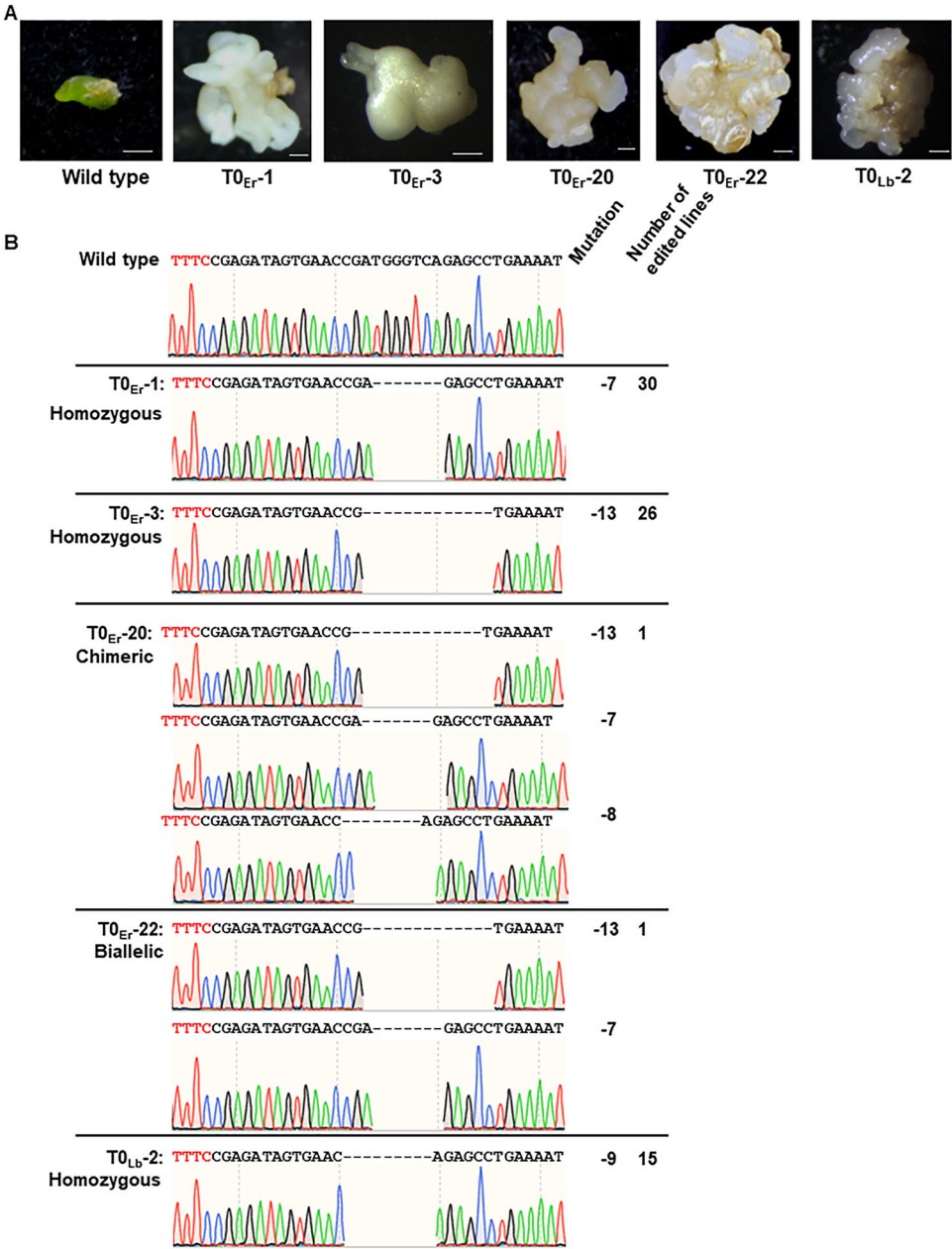

**Fig. 3 | Genome editing of the *CsPDS* gene of embryogenic protoplast cells of *C. sinensis* cv. Hamlin.** A Genome-edited embryos under regeneration. Er indicates ErCas12a. Lb indicates LbCas12aU. **B** Sanger sequencing confirmation of genome-edited lines in the *CsPDS* gene. TTTC in red: PAM (protospacer adjacent motif). Mutation indicates mutations in the edited embryos. The number of edited embryos for each genotype was also shown.

(30 lines) mutants whereas only one line was wild type (Table 1), demonstrating a homozygous/biallelic mutation rate of 97.4%. The edited lines contained 12 different genotypes including 1 homozygous type (−7/−7) and 11 biallelic types (−11/−7, −11/−9, −14/−7, −4/−3, −7/−2, −7/−4, −7/−6, −7/−7 (different), −19/−7, −8/−4, −9/−6+348) (Table 1 and Supplementary Figs. 2–13). Overall, the most common mutation was 7 bp deletion (42 events), followed by −14 (10 events), −4 (7 events), −11 (5 events), −2 (4 events), −3 and −9 (2 events each), −6, −8, −19 and −6+348 (1 event each) (Table 1). The high frequency of 7 bp deletion was consistent with the genotype (−7/−7) of homozygous mutants. We further confirmed the edited lines by conducting whole genome sequencing using next generation sequencing of one representative line for each of the 12 different mutant genotypes (Table 1, Supplementary Table 1, and Supplementary Figs. 2–13). The whole genome sequencing data were in accordance with the Sanger sequencing data and confirmed the biallelic/homozygous mutations for the 12 mutant genotypes (Supplementary Figs. 2–13). Intriguingly, one edited line contained a 348 bp insertion sequence of *C. sinensis* mitochondrial sequence at the target site of one allele of *CsLOB1*. As expected for RNP-mediated genome editing, analyses of whole genome sequencing data demonstrated that all the 12 edited lines did not contain foreign genes.

We investigated whether our transgene-free lines contained off-target mutations. We searched for potential off-target sites of the crRNA targeting *CsLOB1* gene using the CRISPR P v2.0 program and only one homologous site that differed by up to 4 nucleotides was identified. Both amplicon deep sequencing (Supplementary Table 2) and whole genome sequencing analyses showed no off-target mutations. In addition, we further investigated whether mutations occurred in *CsLOB1* homologs. *C. sinensis* contains two *CsLOB1* functional homologs, *CsLOB2* and *CsLOB3* that share 67.9% and 71.0% identities to *CsLOB1*, respectively[50]. Whole genome sequencing analysis showed

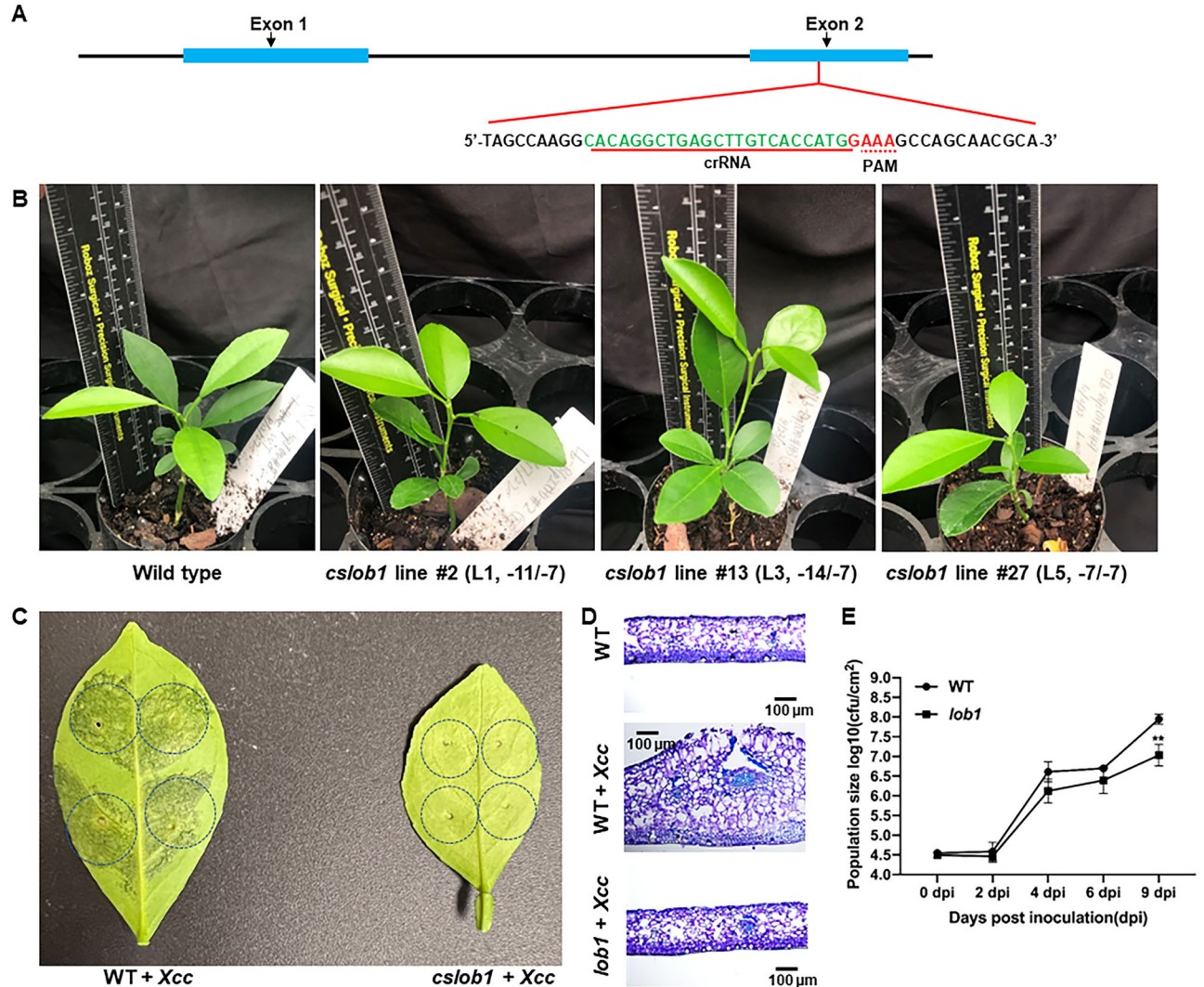

**Fig. 4 | Transgene-free *cslob1* mutants of *Citrus sinensis* cv. Hamlin generated by genome editing of the *CsLOB1* gene. A** Schematic representation of the *CsLOB1* gene and crRNA. Blocks in blue indicate exons. Line fragments indicate introns. Nucleotides in green: crRNA. GAAA in red: PAM. **B** Representative transgene-free *cslob1* mutants of *C. sinensis* cv. Hamlin grafted on Carrizo citrange (*Poncirus trifoliata × Citrus sinensis*) kept in greenhouse. The genotypes of the mutants were demonstrated. Wild type Hamlin generated from seeds was grafted on the same rootstock. **C** Canker symptoms on wild type *C. sinensis* cv. Hamlin and *cslob1* mutant. Fully expanded citrus leaves were inoculated with *Xcc* at $10^7$ CFU/mL using needleless syringes. The picture was taken at 9 days after inoculation. Six biological replicates were tested with similar results. Only one representative picture was shown. **D** Thin cross-section images of **C. E** *Xcc* titers at 9 days post-inoculation. Three biological replicates were used and mean values ± SD (*n* = 3) are shown. The experiments were repeated at least two times with comparable results. Student's *t* test was used for statistical analysis, double asterisks (**) showed significant differences (*P* value < 0.05). Source data are provided as a Source data file.

that *CsLOB2* and *CsLOB3* sequences in the 12 edited lines were identical to their counterparts in wild type *C. sinensis* cv. Hamlin.

### Evaluation of *cslob1* mutants

Among the 38 biallelic/homozygous *cslob1* mutants 32 lines were similar as wild type *C. sinensis* cv. Hamlin in growth phenotypes. However, 6 lines showed narrower leaves (Fig. 4B and Supplementary Fig. 14). Because the majority of the regenerated lines had similar leaf phenotypes as wild type plants, it is probably the narrow leaf phenotype of the 6 regenerated lines resulted from somaclonal variation in tissue culture. As expected, *Xcc* infection of the biallelic/homozygous *cslob1* mutants did not cause any canker symptoms (Fig. 4C and Supplementary Fig. 15). The typical hypertrophy and hyperplasia in leaf tissues caused by *Xcc* were abolished by mutation of the *CsLOB1* gene (Fig. 4D). Significant differences in *Xcc* titers were observed between the wild type and *cslob1* mutants (Fig. 4E). In addition, we also conducted foliar spray of wild type and *cslob1* mutants with *Xcc* to mimic the natural infection of *Xcc*. Canker symptoms were observed around

the wounds of wild type Hamlin, but not that of *cslob1* mutants. Consistently, *Xcc* titers were significantly lower in the *cslob1* mutants than the wild type Hamlin (Fig. 5).

To explore the canker resistance mechanism of the *cslob1* mutants, we investigated the expression of *Cs7g32410* (expansin), *orange.1t00600* (3-oxo-5-alpha-steroid 4-dehydrogenase), *Cs6g17190* (RSI-1), and *Cs9g17380* (PAR1), which were known to be up-regulated by CsLOB1 during *Xcc* infection[50–52], in the wild type and *cslob1* mutant. Quantitative reverse-transcription PCR (qRT-PCR) analysis clearly demonstrated that expression of *orange.1t00600*, *Cs6g17190*, *Cs7g32410* and *Cs9g17380* was significantly lower in the *cslob1* mutant than in the wild type *C. sinensis* in the presence of *Xcc* (Fig. 6A). Thus mutation of *CsLOB1* abolished the induction of downstream genes of *CsLOB1* by *Xcc*, which explains the obliteration of canker symptoms on the *cslob1* mutant, consistent with previous studies[50,51,53]. In addition, reactive oxygen species (ROS) production is known to play critical roles in suppressing pathogen growth in plants[54,55]. The concentrations of $H_2O_2$, an indicator of ROS, were similar among wild type and the

*cslob1* mutant of *C. sinensis* with or without *Xcc* inoculation at 1 day post inoculation (DPI) (Fig. 6B). This is consistent with *C. sinensis* being susceptible to *Xcc* and there is no significant ROS induction in *C. sinensis* by *Xcc*[56].

## Discussion

In this study, we have generated transgene-free canker-resistant *C. sinensis* cv. Hamlin lines via Cas12a/crRNA RNP transformation of embryogenic citrus protoplasts by editing the coding region of canker susceptibility gene *CsLOB1*. Canker resistance resulting from editing the coding region of canker susceptibility *CsLOB1* is consistent with previous results in enabling canker resistance by editing the promoter region or coding regions of *LOB1* genes in grapefruit (*C. paradisi*)[27,29], sweet orange (*C. sinensis*)[31,32] and pummelo (*C. maxima*)[28]. Interestingly, natural variations of the effector binding elements in the *LOB1* promoter were reported to contribute to citrus canker disease resistance in *Atalantia buxifolia*, a primitive (distant-wild) citrus[57]. Mutation

**Table 1 | Summary of transgene-free *CsLOB1*-edited *C. sinensis* cv. Hamlin lines generated by LbCas12aU/crRNA RNP transformation of embryogenic citrus protoplasts**

| Types of regenerated lines | Mutation types | Mutations of two alleles | Number of regenerated lines |
|---|---|---|---|
| L1 | Biallelic | −11/−7 | 4 |
| L2 | Biallelic | −7/−2 | 4 |
| L3 | Biallelic | −14/−7 | 10 |
| L4 | Biallelic | −19/−7 | 1 |
| L5 | Homozygous | −7/−7 | 8 |
| L6 | Biallelic | −7/−4 | 4 |
| L7 | Biallelic | −7/−7 (different) | 1 |
| L8 | Biallelic | −8/−4 | 1 |
| L9 | Biallelic | −7/−6 | 1 |
| L10 | Biallelic | −4/−3 | 2 |
| L11 | Biallelic | −11/−9 | 1 |
| L12 | Biallelic | −9/−6+348 | 1 |
| L13 | Wild type | | 1 |

of the coding region or promoter region of susceptibility genes via genome editing or utilizing their natural variants has been successfully used in generating disease-resistant plants such as bacterial blight-resistant rice varieties[58], powdery mildew-resistant wheat[59], and enabling broad resistance to bacterial, oomycete, and fungal pathogens[60].

Biotechnological approaches including transgenic expression, RNAi, and CRISPR mediated genome editing have been used in citrus genetic improvement[13,16,17,19,20,61–64]. However, none of them have been adopted for commercial use despite significant improvements in different traits including resistance to diseases and shortened juvenility. The lack of success in commercialization for citrus plants generated by biotechnological approaches primarily results from their transgenic nature. Transgenic plants need to pass rigorous, lengthy, and costly regulatory approvals. The regulatory requirements by different countries/regions vary[33,34]. In the U.S., transgenic plants are regulated by the Animal & Plant Health Inspection Service (APHIS), Environmental Protection Agency (EPA), and Food and Drug Administration (FDA). Our *CsLOB1* edited *C. sinensis* cv. Hamlin lines were generated by the RNP method that does not involve DNA fragments. Consequently, the edited *C. sinensis* lines do not contain foreign genes, which is consistent with other genome-edited plants generated by the RNP approach[36–41]. In agreement with the low off-target efficacy of RNP-mediated genome editing[39,40], the *CsLOB1* edited *C. sinensis* cv. Hamlin lines do not have off-target mutations including in *CsLOB2* and *CsLOB3*, the two *CsLOB1* homologs. Importantly, most of the *CsLOB1* edited *C. sinensis* cv. Hamlin lines demonstrate no phenotypic differences from wild type plants except canker resistance. However, 6 regenerated lines showed slightly narrower leaves than the wild type and other edited lines, which might result from somaclonal variation in tissue culture[65]. Mutation of the *LOB1* homolog in *Arabidopsis* was also reported to have no effect on plant phenotypes due to functional redundancy of *LOB1* gene and its homologs[66]. Owing to the long juvenile period, we were unable to evaluate the fruit quality and yield of the edited lines, which is expected to be completed in 3 more years. Because of the aforementioned traits, the transgene-free canker-resistant *C. sinensis* cv. Hamlin lines have received regulatory approval by APHIS, clearing one important hurdle for its potential use in production and exempted from regulation by EPA.

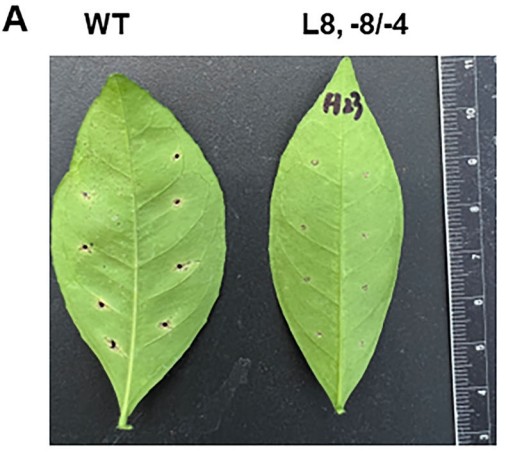

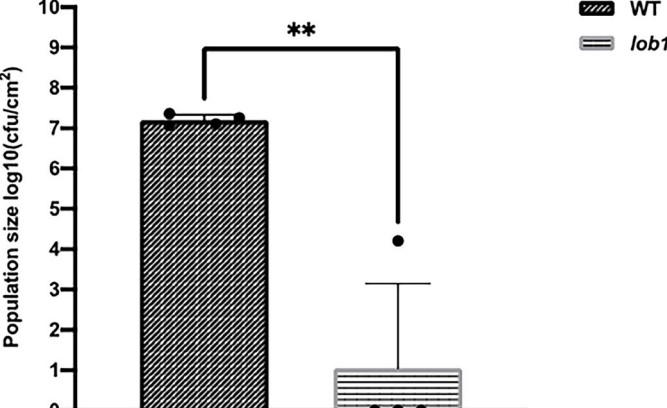

**Fig. 5 | Symptoms of wild type and transgene-free *cslob1* mutants of *Citrus sinensis* cv. Hamlin after foliar spray with *Xanthomonas citri* subsp. citri.**
**A** Canker symptoms on wild type *C. sinensis* cv. Hamlin and *cslob1* mutant. Fully expanded citrus leaves were punctured with syringes to make 8 wounds/leaves, followed by foliar spray with *Xcc* at 5 × 10⁸ CFU/mL. The sprayed plants were covered with plastic bag to keep humidity to facilitate infection. The picture was taken at 18 days after inoculation. **B** *Xcc* titers at 18 days post-inoculation. Four biological replicates were used and mean values ± SD (*n* = 4) are shown. The experiments were repeated at least two times with comparable results. Two-sided Student's *t* test was used for statistical analysis, double asterisks (**) showed significant differences (*P* value = 0.00112). Source data are provided as a Source data file.

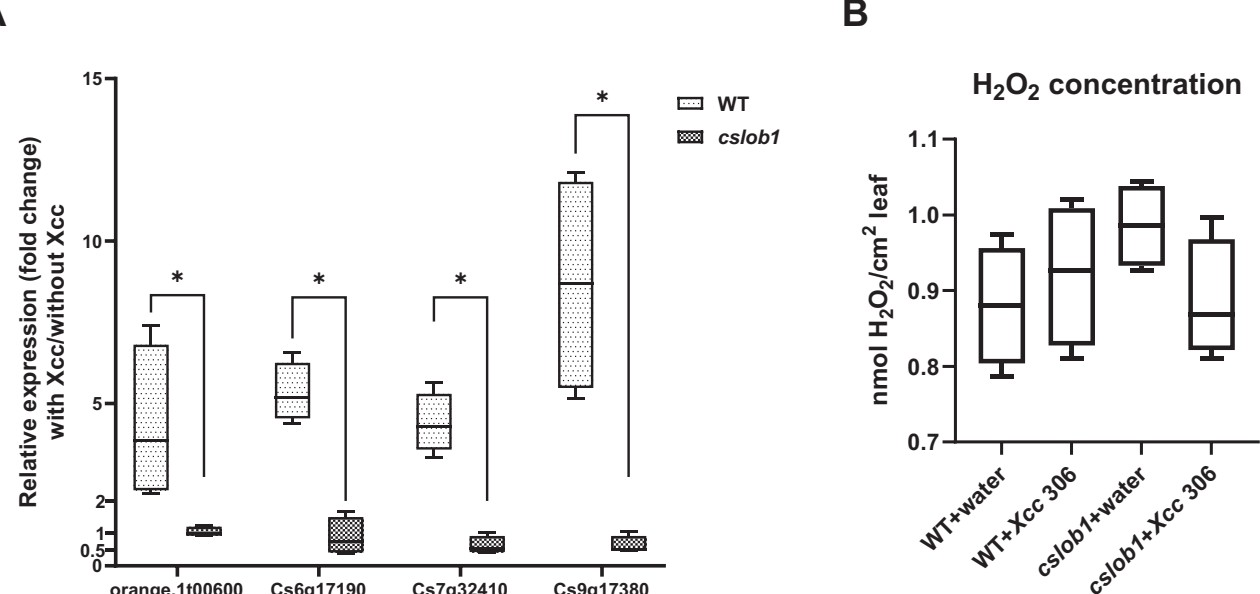

**Fig. 6 | Effect of mutation of *CsLOB1* on ROS production and expression of cell wall related genes. A** Expression of *orange.1t00600*, *Cs6g17190*, *Cs7g32410* and *Cs9g17380*, which are known to be up-regulated by CsLOB1 during *Xcc* infection, in the *cslob1* mutant and wild-type *C. sinensis* cv. Hamlin with and without *Xcc* inoculation at $1 \times 10^7$ cfu/mL with syringes. *CsGAPDH*, a housekeeping gene encoding glyceraldehyde-3-phosphate dehydrogenase in citrus was used as an endogenous control. Four biological replicates were used and mean values ± SD (standard deviation) ($n = 4$) are shown. Two-sided Student's *t* test was used for statistical analysis, single asterisk (*) showed significant differences (*P* value of *orange.1t00600*, *Cs6g17190*, *Cs7g32410* and *Cs9g17380* are 0.0338, 0.000408,

0.000265 and 0.000325, respectively). Experiments were repeated at least two times with similar results and representative results are shown. **B** $H_2O_2$ concentration was quantified one day after inoculation. Four biological replicates were used for each experiment. Wild type and *cslob1* mutant of *C. sinensis* leaves were inoculated with *Xcc* ($1 \times 10^8$ cfu/mL) or water using needleless syringes. Values represent means ± SD ($n = 4$). The experiment was repeated twice with similar results. All box plots include the median line, the box denotes the interquartile range (IQR), whiskers denote the rest of the data distribution and there are no outliers. The lower and upper hinges of the boxes correspond to the 25th and 75th percentiles. Source data are provided as a Source data file.

Canker resistance via editing of *LOB1* gene, the canker susceptibility gene, is demonstrated in multiple aspects. Editing of *LOB1* gene significantly reduces *Xcc* growth when inoculated through injection. In addition, it also abolishes canker symptom development, consistent with obliterating the gene induction of downstream genes of *CsLOB1*. However, mutation of *LOB1* gene has no significant effect on ROS levels, which are known to contribute to suppress pathogen growth in plants[67], suggesting an unknown disease resistance mechanism which is yet to be investigated.

In our study, 38 of the 39 regenerated *CsLOB1* edited lines were biallelic/homozygous mutants, demonstrating a 97.4% biallelic/homozygous mutation rate, which was unexpectedly high considering that it took us approximately 6 years to finally figure out how to generate the transgene-free *CsLOB1* edited *C. sinensis*. In previous studies, the mutation efficacy of RNP-mediated genome editing of protoplast using Cas/gRNA varies with 0.85–5.85% in maize[68], 18% in oil palm[69], 11.9–14.7% in carrot[70], 46.7% in sorghum[71], nearly 100% in rice and 90.8% in citrus[72], and up to 100% in both rice and poplar plants in T0 generation[73]. Thus, our high biallelic/homozygous editing efficacy is not totally unexpected. The different mutation rates for *PDS* and *LOB1* genes and different batches of embryogenic *C. sinensis* protoplasts suggest optimization of crRNA selection and protoplasts is critical for transgene-free genome editing of citrus via the RNP method. In addition, it is probable that mutation of *LOB1* might help protoplast regeneration. It remains to be determined whether such a high editing efficacy can be achieved for other citrus genes beyond *LOB1*. The high efficacy of RNP-mediated citrus genome editing indicates room for improvement and optimization for RNP genome editing in other plant species with low efficacy. The entire process of RNP-mediated citrus genome editing, from transformation to grafting, takes about 10 months (Fig. 1), thus complementing traditional citrus breeding approaches.

In sum, this study generated transgene-free canker-resistant *C. sinensis* lines that are in the process of being evaluated and released to provide a sustainable and efficient solution to control citrus canker, a major plant disease. The efficient transgene-free genome editing approach for citrus using RNP is anticipated to have a significant impact on the genetic improvement of elite citrus cultivars.

## Methods
### Growth conditions of citrus plants and cell culture
For *C. sinensis*, the young seedlings were grown in a greenhouse located in Citrus Research and Education Center, Lake Alfred, FL. Embryogenic callus lines of *C. sinensis* (Hamlin sweet orange) were initiated from immature ovules and maintained on Murashige and Tucker (1969, MT) medium (M5525, PhytoTech Labs, Lenexa, KS, USA)[74] supplemented with 5.0 mg/l Kinetin (KIN) and 500 mg/l malt extract. The suspension cell culture of *C. sinensis* cv. Hamlin was maintained under dark at 22 °C and sub-cultured every two weeks. The growing medium was H + H medium (MT basal medium plus 35 g/L sucrose, 0.5 g/L malt extract, 1.55 g/L glutamine, pH 5.8)[75]. At 7–10 days after subculturing, the suspension cells were used for protoplast isolation.

### Protoplast isolation
Embryogenic *C. sinensis* cv. Hamlin protoplasts were isolated from the suspension cells after digestion with digestion solution (2.5× volume BH3 and 1.5× volume enzyme solution (0.7 M mannitol, 24 mM CaCl₂, 6.15 mM MES buffer, 2.4 % (w/v) Cellulase Onozuka RS (MX7353, Yakult Honsha, Minato-ku, Tokyo, Japan), 2.4 % (w/v) Macerozyme R-10 (MX7351, Yakult Honsha), pH 5.6) for 16-20 hours at 28 °C. After digestion, the digestion protoplast mixture was filtered with a 40 μM cell strainer (431750, Corning, Durham, NC, USA) into a 50 mL Falcon tube, which were centrifuged at 60 g for 7 min. The pellets were

resuspended with BH3 medium to wash the protoplast[48]. After repeating the washing step, the protoplasts were resuspended in 2 mL BH3 medium and diluted to $1 \times 10^6$ cell/mL and kept in dark at room temperature for 1 hour.

## Cas12a proteins and crRNA molecules

ErCas12a protein with a single, carboxy-terminal SV40-derived nuclear localization signal was received from Integrated DNA Technologies (IDT, Coralville, IA). DNA sequence encoding ErCas12a was cloned into the pET28a vector by Gibson assembly. For protein expression, a single transformed *E. coli* BL21(DE3) colony was inoculated into LB medium supplemented with Kanamycin (25 μg/mL), and grown overnight at 37 °C, 250 rpm. The overnight culture was transferred to terrific broth medium containing 0.5% glucose and 25 μg/mL kanamycin, grown at 37 °C, 250 rpm for -2–3 h until $OD_{600}$ reached 0.6. The culture was chilled at 4 °C for 30 min prior to induction with 1 mM IPTG, and further incubated at 18 °C, 250 rpm for 12–18 h. The recombinant ErCas12a protein was purified as previously described[76]. Briefly, *E. coli* cells were harvested by centrifugation, and homogenized with Emulsiflex-C3 high-pressure homogenizer (Avestin, Ottawa ON, Canada). The ErCas12a protein in clarified lysate was sequentially purified using immobilized metal affinity chromatography (HisTrap HP, GE Healthcare) and heparin chromatography (HiTrap Heparin HP, GE Healthcare). Purified protein was concentrated and dialyzed against storage buffer (20 mM TrisHCl, 300 mM NaCl, 0.1 mM EDTA, 50% glycerol, 1 mM DTT, pH 7.4) overnight. The protein concentration was measured by Nano-Drop using extinction coefficient at 143,940 $M^{-1} cm^{-1}$, diluted to 60 μM, and stored at −20 °C. Alt-RL.b. Cas12a (Cpf1) Ultra (LbCas12aU) protein with a single carboxy-terminal SV40-derived nuclear localization signal was purchased from IDT (Catalog#: 10007924). crRNAs targeting *CsPDS* or *CsLOB1* genes were selected by manually searching for the PAM site (TTTV). crRNAs (Supplementary Table 2) targeting *CsPDS* or *CsLOB1* genes were synthesized by IDT and diluted to 0.05 nmol/μL by RNase-free water.

## In vitro digestion

ErCas12a (1 μg) or LbCas12aU (1 μg) protein and 1 μg crRNA were assembled in 1X Nuclease Reaction Buffer (B6003S, New England BioLabs, Ipswich, MA, USA) at room temperature for 10 minutes. Then 100 ng DNA fragments were added to the mixture in a total volume of 30 μL. Digested DNA products were run using 2% agarose gel after 30 min digestion at 37 °C.

## Transformation of embryogenic citrus protoplast and plant regeneration

For RNP assembly, 0.27 nmol ErCas12a/LbCas12aU protein and 0.45 nmol crRNA were assembled in 1× Nuclease Reaction Buffer (NEB). The protein and RNA were mixed and incubated for 10 minutes at room temperature and used for transformation of embryogenic *C. sinensis* cv. Hamlin protoplasts using the PEG method[48].

For each transfection reaction, 1 mL protoplast cells, 20 μL pre-assembled RNP, and 1 mL PEG-CaCl₂ (0.4 M mannitol, 100 mM CaCl₂, and 40% PEG-4000) were mixed and kept at room temperature for 15 min in dark followed by washing with BH3 medium twice. Then the edited protoplasts were resuspended by 1 mL of a 1:1 (v:v) mixture of BH3 0.6 M and EME sucrose 0.6 M liquid medium. The RNP-transformed embryogenic citrus protoplasts were used for plant regeneration[48] (Fig. 1).

EME sucrose 0.6 M liquid medium: 4.46 g/L Murashige & Tucker Medium (M5525, PhytoTech Labs), 205.4 g/L sucrose, 0.5 g/L malt extract, pH 5.8; filter-sterilize and store at room temperature.

EME sucrose 0.15 M liquid medium: 4.46 g/L Murashige & Tucker Medium, 50 g/L sucrose, 0.5 g/L malt extract, pH 5.8; filter-sterilize and store at room temperature.

EME sucrose solid medium: 4.46 g/L Murashige & Tucker Medium, 50 g/L sucrose, 0.5 g/L malt extract, 3.2 g/L Gelzan (G3251, PhytoTech Labs), pH 5.8; autoclave medium and pour into 100 × 25 mm petri dishes.

EME maltose 0.15 M liquid medium: 4.46 g/L Murashige & Tucker Medium, 50 g/L maltose, 0.5 g/L malt extract, pH 5.8; filter-sterilize and store at room temperature.

EME maltose solid medium: 4.46 g/L Murashige & Tucker Medium, 50 g/L maltose, 0.5 g/L malt extract, 3.2 g/L Gelzan, pH 5.8; autoclave medium and pour into 100 × 25 mm petri dishes.

EME1500 solid medium: 4.46 g/L Murashige & Tucker Medium, 50 g/L maltose, 1.5 g/L malt extract, pH 5.8, 3.2 g/L Gelzan; autoclave medium and pour into 100 × 25 mm petri dishes.

MS medium: 34.43 g/L Murashige and Skoog Basal Medium with vitamins and sucrose (M5501, PhytoTech Labs), pH 5.8, 3.2 g/L Gelzan; autoclave medium and pour into 25 × 150 mm glass tubes.

Liquid rooting media: 4.46 g/L Murashige & Tucker Medium, 30 g/L sucrose, 200 μg/L NAA, 30 μg/L IBA, pH 5.8; autoclave and pour into 25 ×150 mm glass tubes stored at room temperature.

BH3 medium: 10 mL/L BH3 macronutrient stock, 10 mL/L MT micronutrient stock, 10 mL/L MT vitamin stock, 15 mL/L MT calcium stock, 5 mL/L MT iron stock, 2 mL/L BH3 multivitamin stock A, 1 mL/L BH3 multivitamin stock B, 1 mL/L BH3 KI stock, 10 mL/L BH3 sugar and sugar alcohol stock, 20 mL/L BH3 organic acid stock, 20 mL/L coconut water, 82 g/L mannitol, 51.3 g/L sucrose, 3.1 g/L glutamine, 1 g/L malt extract, 0.25 g/L casein enzyme hydrolysate, pH 5.8; filter-sterilize and store at room temperature.

BH3 macronutrient stock: 150 g/L KCl, 37 g/L MgSO₄·7H₂O, 15 g/L KH₂PO₄, 2 g/L K₂HPO₄; dissolve in H₂O and store at 4 °C.

MT micronutrient stock: 0.62 g/L H₃BO₃, 1.68 g/L MnSO₄·H₂O, 0.86 g/L ZnSO₄·7H₂O, 0.083 g/L KI, 0.025 g/L Na₂MoO₄·2H₂O, 0.0025 g/L CuSO₄·5H₂O, 0.0025 g/L CoCl₂·6H₂O; dissolve in H₂O and store at 4 °C.

MT vitamin stock: 10 g/L myoinositol, 1 g/L thiamine-HCl, 1 g/L pyridoxine-HCl, 0.5 g/L nicotinic acid, 0.2 g/L glycine; dissolve in H₂O and store at 4 °C.

MT calcium stock: 29.33 g/L CaCl₂·2H₂O; dissolve in H₂O and store at 4 °C.

MT iron stock: 7.45 g/L Na₂EDTA, 5.57 g/L FeSO₄·7H₂O; dissolve in H₂O and store at 4 °C.

BH3 multivitamin stock A: 1 g/L ascorbic acid, 0.5 g/L calcium pantothenate, 0.5 g/L choline chloride, 0.2 g/L folic acid, 0.1 g/L riboflavin, 0.01 g/L p-aminobenzoic acid, 0.01 g/L biotin; dissolve in H₂O and store at −20 °C.

BH3 multivitamin stock B: 0.01 g/L retinol dissolved in a few drops of alcohol, 0.01 g/L cholecalciferol dissolved in a few drops of ethanol, 0.02 g/L vitamin B12; dissolve in H₂O and store at −20 °C.

BH3 KI stock: 0.83 g/L KI; dissolve in H₂O and store at 4 °C.

BH3 sugar and sugar alcohol stock: 25 g/L fructose, 25 g/L ribose, 25 g/L xylose, 25 g/L mannose, 25 g/L rhamnose, 25 g/L cellobiose, 25 g/L galactose, 25 g/L mannitol; dissolve in H₂O and store at −20 °C.

BH3 organic acid stock: 2 g/L fumaric acid, 2 g/L citric acid, 2 g/L malic acid, 1 g/L pyruvic acid; dissolve in H₂O and store at −20 °C.

## Mutation detection

Genomic DNA was extracted from leaves of wild type or *cslob1* mutants of *C. sinensis* cv. Hamlin. For *cspds* mutants, genomic DNA was extracted from embryos. Primers used for PCR were listed in the Supplementary Table 2. CloneAmp HiFi PCR Premix (639298, Takara Bio USA, San Jose, CA, USA) was used for PCR amplification following the manufacturer's instructions using the following protocol: 98 °C for 30 s; followed by 40 cycles at 98 °C for 10 s, 54 °C for 10 s, and 72 °C for 45 s; followed by a final extension at 72 °C for 5 min. PCR amplicons were sequenced directly using the amplifying primers or cloned with Zero Blunt TOPO PCR Cloning Kit (450245, Thermo Fisher, San Jose,

CA, USA) and transformed into Stellar Competent Cells (Takara). M13-F (GTAAAACGACGGCCAGTG) and M13-R (CAGGAAACAGCTATGACC) were used for single colony PCR amplification and Sanger sequencing. For deep sequencing, PCR amplicons of all the mutants were purified and mixed, then sent to Genewiz (South Plainfield, NJ, USA) for the next-generation sequence (2 ×250 bp paired-end reads). Data were analyzed by the online tool Cas-Analyzer[1] for mutation detection.

DNA Library construction, sequencing, and data analysis. Following the manufacturer's protocol of short read DNA sequencing from Illumina[77], the library was prepared. After quality control, quantification, and normalization of the DNA libraries, 150 bp paired-end reads were generated using the Illumina NovaSeq 6000 platform according to the manufacturer's instructions at Novogene. The raw paired-end reads were filtered to remove low-quality reads using fastp program version 0.22.0[78]. On average, more than 21.45 Gb of high-quality data was generated for each edited sweet orange plant sample (Supplementary Table 1). To identify the mutations (single nucleotide polymorphisms, deletions and insertions) for the mutated plant genomes, the high-quality paired-end short genomic reads were mapped to sweet orange (*C. sinensis*)[79] reference genome using Bowtie2 software version 2.2.6[80]. Based on the mapping results, mutations were detected using the SAMtools package version 1.2[81] and deepvariant program version 1.4.0[82]. The generated mutations were filtered by quality and sequence depth (mapping quality >10 and mapping depth >10). The mutations of target site were visualized using the Integrative Genomics Viewer (IGV) software version 2.15.4[83]. The high-quality paired-end short reads were further used to detect foreign DNA sequences. The off-target sites were predicted by using CRISPR-P 2.0 program[49] and aligning target sequence with whole genome using blast program. Based on the mapping results, mutations of off-target sites were detected using the SAMtools package version 1.2 and deepvariant program version 1.4.0.

### Quantitative reverse-transcription PCR
*Xcc* strain 306 was infiltrated into wild type *C. sinensis* cv. Hamlin and transgene-free *cslob1* mutants at the concentration of $1 \times 10^7$ cfu/mL for cell wall related genes. The infiltration-area of the leaf samples were collected at 9 days post-inoculation (dpi) for RNA isolation for cell wall related genes. Four biological repeats were used with one leaf as one biological replicate. Total RNA was extracted by TRIzol Reagent (15596026, Thermo-Fisher) following the manufacturer's instructions. cDNA was synthesized by qScript cDNA SuperMix (101414, Quantabio, Beverly, MA, USA). Primers used for qRT-PCR were listed in Supplementary Table 2. Briefly, the real-time PCR was performed with QuantiStudio3 (Thermo-Fisher) using SYBR Green Real-Time PCR Master Mix (4309155, Thermo-Fisher) in a 10 μL reaction. The standard amplification protocol was 95 °C for 3 min followed by 40 cycles of 95 °C 15 s, 60 °C for 60 s. *CsGAPDH* was used as an endogenous control. All reactions were performed in triplicate. Relative gene expression and statistical analysis were calculated using the $2^{-\Delta\Delta CT}$ method[84]. qRT-PCR was repeated twice with similar results.

### Microscopy analysis
The infiltration areas of *Xcc*-infiltrated wild type *C. sinensis* cv. Hamlin and *cslob1* mutant leaves and non-inoculated wild type Hamlin leaves were cut with sterilized blades and fixed in 4% paraformaldehyde for at least 2 h. The specimen was dehydrated and embedded in paraffin chips. The paraffin chips were sectioned with Leica 2155 microtome and the thickness of the cut ribbon was 8 μm. The ribbons were located on the glass slides and incubated at 37 °C overnight to be heat fixed. Followed by the dewaxing and rehydrating process, the slides were stained with 0.05% Toluidine blue for 30 s, then rinsed in ddH$_2$O, dehydrated, and added one drop of mounting media, covered with a coverslip. After solidifying for 1 h, the photos of the slides were taken with Leica LasX software (Leica Biosystems Inc.,

Lincolnshire, IL, USA) under the bright-field microscope (Olympus BX61; Olympus Corporation, Shinjuku City, Tokyo, Japan).

### *Xcc* growth assay
Leaf disks (0.5 cm in diameter) punched from the inoculated plant leaves were ground in 0.2 mL sterilized H$_2$O. 100 μL serial dilutions of the grinding suspensions were spread on NA plates (dilutions ranging from $10^{-1}$ to $10^{-6}$). Bacterial colonies were counted after 48 h and the number of CFU per cm$^2$ of leaf disc was calculated and presented with Prism GraphPad software.

### Quantification of H$_2$O$_2$ concentration
H$_2$O$_2$ concentration measurement was conducted according to previously described method[56]. Briefly, four 0.5-cm-diameter leaf disks from the same leaf that had been injected with water or *Xcc* ($1 \times 10^8$ cfu/mL) were pooled and stored in a 1.5 mL tube with 0.5 mL of double-distilled (DD) water. The samples were rotated on a platform at 20 rpm for 30 min, and DD water was replenished with fresh DD water. Samples were incubated for an additional 6 h on a rotating platform at 20 rpm. H$_2$O$_2$ concentration was measured in the supernatants using the Pierce Quantification Peroxide Assay Kit (23280, Thermo Fisher Scientific, Waltham, MA, USA).

### Reporting summary
Further information on research design is available in the Nature Portfolio Reporting Summary linked to this article.

## Data availability
The raw reads of genome resequencing for sweet orange plants generated in this study were deposited in the NCBI Bioproject database under the accession number PRJNA931574. The reference genome of sweet orange was downloaded from Citrus Pan-genome to Breeding Database [http://citrus.hzau.edu.cn/index.php]. Source data are provided with this paper.

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

## Acknowledgements

We thank Wang lab members for constructive suggestions and insightful discussions. This project was supported by funding from Florida Citrus Initiative Program, Citrus Research and Development Foundation 18-025, U.S. Department of Agriculture National Institute of Food and Agriculture grants 2022-70029-38471, 2021-67013-34588, 2018-70016-27412 and 2016-70016-24833, FDACS Specialty Crop Block Grant Program AM22SCBPFL1125 (N.W.).

## Author contributions

N.W. conceptualized, designed the experiments and supervised the project. H.S. and Y.W. performed the experiments. A.A.O. and J.W.G. provided citrus suspension culture, L.Z. and C.A.V. provided Cas12a proteins. J.X. and H.S. performed bioinformatics and statistical analyses. N.W., H.S., J.X., and Y.W. wrote the manuscript with input from all co-authors.

## Competing interests

N.W., H.S. and Y.W. filed a patent (application number: UFTT19061) based on the results reported in this paper. The patent application is about transgene-free genome editing of citrus using Cas12a/crRNA ribonucleoprotein and edited site in the coding region of *LOB1* gene for canker resistance. All other authors declare no competing interests.
