## [Peer Review File · Nature Communications]

Generation of the transgene-free canker-resistant *Citrus sinensis* using Cas12a/crRNA ribonucleoprotein in the T0 generationReviewers' Comments:

Reviewer #1:

Remarks to the Author:

The authors report the production of gene edited T0 Citrus sinensis lines with mutations in the coding region of the LOB1 gene. The produced lines are reported as resistant to citrus canker and contain no off-target mutations.

My main concern relates with the supposed efficiency of the method. This work claims that 97.4% of the recovered T0 lines contain homozygous/bi-allelic mutations in the target gene, at least for LOB1 (they do not show efficiency percentages for the experiments targeting PDS1).

The work claims that after PEG-mediated transfection of protoplasts and recovery of plantlets without any selection, 38 out of 39 plantlets have homozygous/bi-allelic edits.

For this to happen:

1.- protoplast transfection efficiency should have been pretty much 100% (something unlikely)

2.- In addition, editing efficiency of transfected protoplasts would need to be almost 100% (something even more unlikely)

That adds up to two consecutive 100% efficiencies.

The discussion section mentions the 97.4% efficiency but does not elaborate on how this almost impossible efficiency in transfection and editing has been achieved.

I am not accusing the authors of misrepresenting the truth or lying, but I just can't believe these numbers.

Other comments.

Line 28

"The CsLOB1 edited C. sinensis lines demonstrate no differences from wild type plants except canker resistance."

These lines do not possess resistance to canker since they exhibit the same bacterial levels in their tissues. These lines are tolerant since they do not show disease symptoms but not resistant.

Resistance the capacity of a plant to inhibit pathogen infection.

Lines 93-108

Preliminary experiments were performed targeting the PDS gene as mutations in this gene result in an easily identifiable albino phenotype. 58 albino embryos were analyzed for transformation with ErCas12a and 15 albino embryos analyzed for LbCas12aU.

I would like to see the overall editing efficiency in these experiments. i.e. How many embryos were obtained in each of the above mentioned experiments and how many of them had an albino phenotype.

Line 116

"Because LbCas12aU demonstrated superior activity in in vitro digestion of target sequence (Fig. 1) and its high efficacy in genome editing of the CsPDS gene (Fig. 2), we used LbCas12aU/crRNA RNP in downstream studies.."

Efficiency of PDS experiments is not shown in the manuscript.

Reviewer #2:

Remarks to the Author:

Su et al 2023

In this paper, the authors used gene editing in citrus protoplasts to generate transgene free, T0 plants with edits in the CsLOB1 susceptibility gene which were resistant to the bacterial pathogen Xcc. The

effect of editing CsLOB1 on resistance to Xcc has previously been reported. Using RNPs delivered to citrus protoplasts to achieve transgene free edits has also previously been reported. Thus, the main advance of the current paper is to take these two pieces of the puzzle and stick them together. They characterized the resulting lines for off target edits and resistance to Xcc. TAL-induced up regulation of CsLOB1 and secondary targets were blocked but not bacterial growth, ROS or PR gene induction (except PR2), consistent with the current understanding of this pathosystem and susceptibility target. No off target edits were observed and the resulting plants appeared developmentally normal, at least at the young age analyzed.

Initial work was done by targeting PDS as a rapid way of comparing efficiency of Cas9 and two variants of Cas12a. LbCas12aU worked best, so this was chosen for subsequent studies. This section of the paper felt fairly redundant with an earlier paper published by the same group (10.1111/pbi.13109). If there are important differences between that previous work and what is reported here, I encourage the authors to make this more clear. Next, they targeted the second exon of CsLob1 with RNPs and used the previously published method to re-generate plants from protoplasts. 42 lines were recovered and 39 survived micro-grafting onto another variety. Re-generated plants did not show any differences from wild type. I found this result surprising. With ~40 lines, I would expect at least a couple of off types, simply as a result of somoclonal variation in tissue culture. Can the authors comment on the frequency of off-types more generally?

It is exciting to see this project moving forward and generation of transgene free citrus material that could be distributed to farmers is an impressive achievement. Indeed, the authors include, as a supplemental file, confirmation from USDA that these plants will not be regulated. Despite the importance of the regulatory landscape of gene edited plants, inclusion of such materials in a research paper struck me as a bit odd and I wonder if this aspect of the work would be better suited to a 'perspective/commentary'.

Specific suggested revisions:

1) In Figure 2a, what stages of regeneration is shown? And why do the T0 embryos look so different from wild type (other than just the white color)? In Figure 3, a wildtype plant is shown for comparison. Is this also a grafted plant? In general, the figure legends could use more details to fully explain what was done.

2) Many claims of 'no developmental defects' are made throughout the manuscript, but this is not supported as the authors acknowledge in the discussion. To evaluate potential effect on fruit/yield would require several more years. I do not advocate that they wait to publish, instead just revise the abstract and results sections text to be more accurate with respect to what phenotypes were evaluated. The authors previously generated (transgenic) gene edited lines. Have these reached maturity and if so, would it be possible to report a more detailed analysis of those lines?

Minor:

Line 40: use of 'dominant' is weird here, haploid organism.

Line 49/50. Is there a reference to support this statement?

Line 84: deletion —> deletions

Line 123: Suggest remove 'Intriguingly.' This is the expected result, albeit, impressive. Line 137 is a good use of 'intriguingly.' But this seems really weird? How would mitochondrial DNA get inserted into the genome?

Line 178: '...did not activate the plant immune system...'

Line 217: Suggest making the discussion of R gene resistance versus susceptibility gene mediated resistance its own paragraph.

Lines 225-228: Really? Interesting. Is there a reference for this?

Lines ~237: I recommend not phrasing this as an alternative to breeding. This is a major advance that will complement ongoing breeding efforts (which may still be more appropriate for some multi genetic

traits and when the causal genes are not known).

Line 243: Similarly, the reference to Huanglongbing feels superficial. Is there a known target that the authors could propose. Yes, that is an important disease, but I would suggest either fleshing this idea out, or excluding it.

Reviewer #3:

Remarks to the Author:

Citrus is an important commodity in the US yet citrus production has been severely challenged by pathogen-induced diseases. Traditional breeding takes long time to achieve while traditional genetic engineering, involving the introduction of foreign DNA fragments into the plant, has met with significant consumer concerns. CRISPR has been highly anticipated as a powerful way to generate transgene-free citrus with improved disease resistance. This report is the first such a study to generate transgene-free citrus plants with improved response to canker disease, using Cas12a/crRNA ribonucleoprotein (RNP)-mediated citrus genome editing. Although the Cas12a/crRNA RNP method is not new and has been used in several other plants, the successful implication of this method in citrus signifies a major advance in generating citrus with improved traits, including disease resistance, development, and more.

The Cas12a/crRNA RNP method was previously reported to have relatively lower efficiency in inducing mutations, compared with the CRISPR/Cas9 method. Based on the data provided in this report, it is unclear though what is the efficiency in citrus. In Fig. 2, albino color conveniently aided the selection of CsPDS-edited embryos. The efficiency of the methods could be roughly derived by examining the ratio of albino and total embryos (both green and albino). The green embryos could also carry heterozygous CRISPR mutations. As a method report, it would be nice to know the overall biallelic frequency on the basis of total frequency (none edited, heterozygous edits, and homozygous edits). Thus, this reviewer suggests the authors to examine more green embryos by sequencing the CsPDS gene and subsequently derive the genome editing efficiency in citrus, using the Cas12a/crRNA RNP method.

Related to the above point, fig. 3 summarizes the editing results for the citrus gene CsLOB1 that confers susceptibility to citrus canker disease. It is unclear how the authors selected the 42 regenerated plantlets, most of which carried biallelic mutations in CsLOB1. The biallelic mutation rate is strikingly high in this report, 97.4% for CsLOB1. Is this high biallelic mutation rate also observed with the CsPDS gene, if both green and albino embryos were counted? If such an observation holds true for both genes, it would suggest that citrus is quite different from other plants in response to the Cas12a/crRNA RNP system. Such a high frequency of obtaining biallelic alleles would make it an ideal plant model of both fundamental studies and practical applications.

Fig. 3: A: more details should be included for the panel, e.g. what do color and underline indicate?

Related to this point, please check other figures and add more details in the figure legends when necessary.

Panel: B-F, images of three edited lines were shown yet the authors described in the legend that the data were similar from six lines. It would be great to show all six lines. The mutant line used to generate data for panels C-F should be indicated.

For panel E, Zou et al (The Plant Journal (2021) 106, 1039–1057) showed that CsLOB1 RNAi plants had reduced Xcc growth in a *C. sinensis* cultivar. However, no difference was observed here between the control plant and the Cslob1 mutants with Xcc infection. Perhaps the authors should do a time course bacterial growth rather than just using one time point.

For Panel F: The authors mentioned that expression of some cell wall genes was up-regulated by CsLOB1 during Xcc infection, in the cslob1 mutant and wild-type *C. sinensis* cv. Hamlin. However,

panel F seems to only show one condition. This reviewer suggests that both conditions, infection and non infection, should be tested for these gene expression.

Fig. 4b: Did the author check gene expression with and without Xcc infection (see comments for Fig. 3F)? Again, please indicate which transgenic line was used.

Line 142: Please rephrase the sentence.

Line 160-166: CsLOB1 was previously known to affect expression of cell wall related genes. Please tone down the importance of the work here – this work did not add to the understanding of canker resistance mechanism mediated by CsLOB1.

Line 264: Two ErCas12a proteins were described. Please indicate clearly which one was used in the related experiments.

Line 299: Plant regeneration should be described in more details. Many abbreviations were used in Fig. S1, making it difficult to know the exact process of plant regeneration. Were there any chemicals used for selecting edited protoplasts and/or embryos?

Line 335-339: It is interesting that different Xcc concentrations and infection times were used for different gene expression analysis. Have the authors tested one Xcc concentration in a time course to see the progression of gene expression for genes of interest?

Responses to reviewers' comment

REVIEWER COMMENTS

Reviewer #1 (Remarks to the Author):

The authors report the production of gene edited TO Citrus sinensis lines with mutations in the coding region of the LOB1 gene. The produced lines are reported as resistant to citrus canker and contain no off-target mutations.

My main concern relates with the supposed efficiency of the method. This work claims that 97.4% of the recovered TO lines contain homozygous/bi-allelic mutations in the target gene, at least for LOB1 (they do not show efficiency percentages for the experiments targeting PDS1).

The work claims that after PEG-mediated transfection of protoplasts and recovery of plantlets without any selection, 38 out of 39 plantlets have homozygous/bi-allelic edits.

For this to happen:

- 1.- protoplast transfection efficiency should have been pretty much 100% (something unlikely)
- 2.- In addition, editing efficiency of transfected protoplasts would need to be almost 100% (something even more unlikely)

That adds up to two consecutive 100% efficiencies.

The discussion section mentions the 97.4% efficiency but does not elaborate on how this almost impossible efficiency in transfection and editing has been achieved.

I am not accusing the authors of misrepresenting the truth or lying, but I just can't believe these numbers.

Response: Actually, we were as surprised as you when we saw the sequencing result for the *LOB1* edited plants.

Please let me describe the process and how we got there.

Transgene-free editing of *LOB1* in citrus was initiated in my lab in 2015. We tested both plasmid and RNP methods for both Cas9/gRNA and Cas12a/crRNA, but did not get any genome edited citrus plants in the first couple of years. We then began to focus on Cas12a and include the PDS gene for editing owing to their obvious albino phenotype. We apologized that we did not calculate the mutation rate for PDS gene because we were really stressed out then that we have not made any biallelic/homozygous mutants in such a long time. We were also told by others that albino phenotype may randomly happen in plants regenerated from protoplasts. With that in mind, we have not included many green embryos in our sequencing analyses for embryos generated from PDS edited protoplasts, but only focused on sequencing the embryos showing albino phenotypes. We did sequencing analysis of RNP transformed protoplasts (3 days after RNP transformation) for PDS editing. We have added the following: PCR amplification and Sanger sequencing analysis of RNP transformed protoplasts at 3 days post transformation (DPT) for *PDS* editing showed 14.3% and 16.7% mutation rate for LbCas12aU/crRNA and ErCas12a/crRNA, respectively.

PCR and sequencing analysis of the PDS gene of the embryos actually showed that all our albino embryos (73 in total) contained mutations as guided by Cas12a/crRNA, instead of random mutations. Anyway, the pds experiment gave us strong hopes to continue even though it was already 2020 then,

over three years since we started this project and even though we still did not get transgene-free *LOB1* edited citrus varieties.

We then focused our effort to edit *LOB1* using Cas12a/crRNA RNA. Sequencing analysis of RNP transformed protoplasts (3 days after RNP transformation) for *LOB1* editing a mutation rate of 48.1% for *LOB1*. Another culture of protoplasts showed a mutation rate of 70% for *LOB1*. Finally, in 2022, we began to successfully generate plants from RNA transformed protoplasts for *LOB1* editing and generated 39 lines (not including 3 died in the process) by the time we submitted the paper and several more afterward. Yes, we were surprised that among the 39 regenerated plants, 38 were homozygous or biallelic mutants.

Here are our thoughts:

1. We routinely obtained a transfection efficiency of approximately 66% or above for protoplasts (Huang X, Wang Y, Xu J, Wang N. Development of multiplex genome editing toolkits for citrus with high efficacy in biallelic and homozygous mutations. *Plant Mol Biol.* 2020 Oct;104(3):297-307).
2. The mutation rate of the plants regenerated from RNP-transformed protoplasts varies for genes, protoplast quality, and gRNA sites.
3. Mutation of *LOB1* might help protoplast regeneration because mutation of *LOB1* seems to slightly increase ROS level.

We have included the following in the manuscript: In our study, 38 of the 39 regenerated *CsLOB1* edited lines were biallelic/homozygous mutants, demonstrating a 97.4% biallelic/homozygous mutation rate, which was unexpectedly high considering that it took us approximately 6 years to finally be able to generate the transgene-free *CsLOB1* edited *C. sinensis*. In previous studies, the mutation efficacy of RNP-mediated genome editing of protoplast using Cas/gRNA varies with 0.85 to 5.85% in maize⁶⁶, 18% in oil palm⁶⁷, 11.9-14.7% in carrot⁶⁸, 46.7% in sorghum⁶⁹, and nearly 100% in rice and 90.8% in citrus⁷⁰. Thus, our high biallelic/homozygous editing efficacy is not totally unexpected. The different mutation rates for *PDS* and *LOB1* genes and different batches of embryogenic *C. sinensis* protoplasts suggest optimization of crRNA selection and protoplasts is critical for transgene-free genome editing of citrus via the RNP method. In addition, it is probable that mutation of *LOB1* might help protoplast regeneration.

Other comments.

Line 28

“The *CsLOB1* edited *C. sinensis* lines demonstrate no differences from wild type plants except canker resistance.”

These lines do not possess resistance to canker since they exhibit the same bacterial levels in their tissues. These lines are tolerant since they do not show disease symptoms but not resistant. Resistance the capacity of a plant to inhibit pathogen infection.

Response: Xcc titer issue was also raised by reviewer 3. During our trouble shooting discussion, we realized that in our original experiment for Xcc growth, Xcc at 10E8 CFU/ml was mistakenly used instead of the regularly used lower concentrations such as 10E7 CFU/ml. We have redone the Xcc titer assays using a time course experiment as suggested by reviewer 3. Xcc titer was significantly reduced at 9 days post inoculation, but only slightly reduced in earlier time points (Fig. 3E).

In addition, we also conducted foliar spray of wild type and *cslob1* mutants with Xcc to mimic the natural infection of Xcc. Canker symptoms were observed around the wounds of wild type Hamlin, but

not that of *cslob1* mutants. Consistently, Xcc titers were significantly lower in the *cslob1* mutants than the wild type Hamlin (Extended Data Fig. S17).

Lines 93-108

Preliminary experiments were performed targeting the PDS gene as mutations in this gene result in an easily identifiable albino phenotype. 58 albino embryos were analyzed for transformation with ErCas12a and 15 albino embryos analyzed for LbCas12aU.

I would like to see the overall editing efficiency in these experiments. i.e. How many embryos were obtained in each of the above mentioned experiments and how many of them had an albino phenotype.

Response: We apologized that we only focused on sequencing the embryos showing albino phenotypes, and did not collect data on green embryos. However, we did PCR amplification and Sanger sequencing analysis of RNP transformed protoplasts at 3 days post transformation (DPT) for *PDS* editing that showed 14.3% and 16.7% mutation rate for LbCas12aU/crRNA and ErCas12a/crRNA, respectively. The information was added in the revision.

Line 116

“Because LbCas12aU demonstrated superior activity in *in vitro* digestion of target sequence (Fig. 1) and its high efficacy in genome editing of the CsPDS gene (Fig. 2), we used LbCas12aU/crRNA RNP in downstream studies..”

Efficiency of PDS experiments is not shown in the manuscript.

Response: We revised this section as follows:

We first tested the mutation rate of the *LOB1* gene for LbCas12aU/crRNA and ErCas12a/crRNA RNPs. PCR amplification and Sanger sequencing analysis of RNP transformed protoplasts at 3 DPT for *LOB1* editing showed 48.1% and 34.8% mutation rate for LbCas12aU/crRNA and ErCas12a/crRNA, respectively. The mutation rate seems to associate with the quality and status of the embryogenic protoplasts because deep sequencing analysis of LbCas12aU/crRNA RNP transformed protoplasts from a different batch at 3 DPT demonstrated 71% mutation rate. In addition, LbCas12aU demonstrated superior activity in *in vitro* digestion of target sequence (Fig. 1).

Reviewer #2 (Remarks to the Author):

Su et al 2023

In this paper, the authors used gene editing in citrus protoplasts to generate transgene free, T0 plants with edits in the CsLOB1 susceptibility gene which were resistant to the bacterial pathogen Xcc. The effect of editing CsLOB1 on resistance to Xcc has previously been reported. Using RNPs delivered to citrus protoplasts to achieve transgene free edits has also previously been reported. Thus, the main advance of the current paper is to take these two pieces of the puzzle and stick them together. They characterized the resulting lines for off target edits and resistance to Xcc. TAL-induced up regulation of CsLOB1 and secondary targets were blocked but not bacterial growth, ROS or PR gene induction (except PR2), consistent with the current understanding of this pathosystem and susceptibility target. No off target edits were observed and the resulting plants appeared developmentally normal, at least at the young age analyzed.

Initial work was done by targeting PDS as a rapid way of comparing efficiency of Cas9 and two variants of Cas12a. LbCas12aU worked best, so this was chosen for subsequent studies. This section of the paper felt fairly redundant with an earlier paper published by the same group (10.1111/pbi.13109). If there are important differences between that previous work and what is reported here, I encourage the authors to make this more clear.

Response: Transgene-free editing of *LOB1* in citrus was initiated in my lab in 2017. We tested both plasmid and RNP methods for both Cas9/gRNA and Cas12a/crRNA, but did not get any genome edited citrus plants in the first 2 years. We then began to focus on Cas12a and include the PDS gene for editing owing to their obvious albino phenotype. In our previous study (10.1111/pbi.13109), binary vector containing LbCas12a was used to modify the *CsPDS* gene in Duncan grapefruit via Xcc-facilitated agroinfiltration/transient expression of leaves. In this study, we used LbCas12aU/crRNA and ErCas12a/crRNA RNPs to transform embryogenic protoplasts with the aim to generate transgene free genome edited citrus.

Next, they targeted the second exon of *CsLob1* with RNPs and used the previously published method to re-generate plants from protoplasts. 42 lines were recovered and 39 survived micro-grafting onto another variety. Re-generated plants did not show any differences from wild type. I found this result surprising. With ~40 lines, I would expect at least a couple of off types, simply as a result of somoclonal variation in tissue culture. Can the authors comment on the frequency of off-types more generally?

Response: Thanks for raising this important point. We have taken pictures of all different genotypes (Extended Data Figure S17). We have revised the manuscript accordingly as follows: Among the 38 biallelic/homozygous *cslob1* mutants 32 lines were similar as wild type *C. sinensis* cv. Hamlin in growth phenotypes. However, 6 lines showed narrower leaves (Fig. 3B, Extended Data Fig. 17). Because the majority of the regenerated lines had similar leaf phenotypes as wild type plants, it is probably the narrow leaf phenotype of the 6 regenerated lines resulted from somaclonal variation in tissue culture.

It is exciting to see this project moving forward and generation of transgene free citrus material that could be distributed to farmers is an impressive achievement. Indeed, the authors include, as a supplemental file, confirmation from USDA that these plants will not be regulated. Despite the importance of the regulatory landscape of gene edited plants, inclusion of such materials in a research paper struck me as a bit odd and I wonder if this aspect of the work would be better suited to a 'perspective/commentary'.

Response: We have removed the file in the revision.

Specific suggested revisions:

1) In Figure 2a, what stages of regeneration is shown? And why do the T0 embryos look so different from wild type (other than just the white color)? In Figure 3, a wildtype plant is shown for comparison. Is this also a grafted plant? In general, the figure legends could use more details to fully explain what was done.

Response: Green embryo was in early stage, approximately 6 months after transformation. PDS mutant embryos were also 6 months after transformation. They failed to regenerate into shoots, but kept expanding. We were unsure whether the failed regeneration resulted from poor quality of protoplasts or our skill.

Wild type was grafted on the same rootstock. Wild type plants were generated from seeds.

2) Many claims of 'no developmental defects' are made throughout the manuscript, but this is not supported as the authors acknowledge in the discussion. To evaluate potential effect on fruit/yield would require several more years. I do not advocate that they wait to publish, instead just revise the abstract and results sections text to be more accurate with respect to what phenotypes were evaluated. The authors previously generated (transgenic) gene edited lines. Have these reached maturity and if so, would it be possible to report a more detailed analysis of those lines?

Response: Agree. Revised accordingly.

Minor:

Line 40: use of 'dominant' is weird here, haploid organism.

Response: deleted.

Line 49/50. Is there a reference to support this statement?

Response: I am aware of several programs in Australia (ongoing) and Florida (in the past) that have done so. We have deleted it.

Line 84: deletion → deletions

Response: revised as suggested.

Line 123: Suggest remove 'Intriguingly.' This is the expected result, albeit, impressive. Line 137 is a good use of 'intriguingly.' But this seems really weird? How would mitochondrial DNA get inserted into the genome?

Response: Deleted the 'Intriguingly' in line 123. We did analysis again. It again shows sequence of Citrus sinensis mitochondrion. We do not understand how it happened.

Line 178: '...did not activate the plant immune system...'

Response: Deleted during revision. Thanks.

Line 217: Suggest making the discussion of R gene resistance versus susceptibility gene mediated resistance its own paragraph.

Response: Agree. Revised as suggested.

Lines 225-228: Really? Interesting. Is there a reference for this?

Response: We have removed this section owing to new data in Xcc titers.

Lines ~237: I recommend not phrasing this as an alternative to breeding. This is a major advance that will complement ongoing breeding efforts (which may still be more appropriate for some multi genetic traits and when the causal genes are not known).

Response: Revised as follows:

The entire process of RNP-mediated citrus genome editing, from transformation to grafting, takes about 10 months (Extended Data Fig. S1), thus complementing traditional citrus breeding approaches.

Line 243: Similarly, the reference to Huanglongbing feels superficial. Is there a known target that the authors could propose. Yes, that is an important disease, but I would suggest either fleshing this idea out, or excluding it.

Response: Removed.

Reviewer #3 (Remarks to the Author):

Citrus is an important commodity in the US yet citrus production has been severely challenged by pathogen-induced diseases. Traditional breeding takes long time to achieve while traditional genetic engineering, involving the introduction of foreign DNA fragments into the plant, has met with significant consumer concerns. CRISPR has been highly anticipated as a powerful way to generate transgene-free citrus with improved disease resistance. This report is the first such a study to generate transgene-free citrus plants with improved response to canker disease, using Cas12a/crRNA ribonucleoprotein (RNP)-mediated citrus genome editing. Although the Cas12a/crRNA RNP method is not new and has been used in several other plants, the successful implication of this method in citrus signifies a major advance in generating citrus with improved traits, including disease resistance, development, and more.

Response: We appreciate the positive comments. It is always encouraging know colleagues appreciate our work considering it took us more than five years to get it done.

The Cas12a/crRNA RNP method was previously reported to have relatively lower efficiency in inducing mutations, compared with the CRISPR/Cas9 method. Based on the data provided in this report, it is unclear though what is the efficiency in citrus. In Fig. 2, albino color conveniently aided the selection of CsPDS-edited embryos. The efficiency of the methods could be roughly derived by examining the ratio of albino and total embryos (both green and albino). The green embryos could also carry heterozygous CRISPR mutations. As a method report, it would be nice to know the overall biallelic frequency on the basis of total frequency (none edited, heterozygous edits, and homozygous edits). Thus, this reviewer suggests the authors to examine more green embryos by sequencing the CsPDS gene and subsequently derive the genome editing efficiency in citrus, using the Cas12a/crRNA RNP method.

Response: Thanks for the excellent suggestions. We did not collect data on green embryos. But we did collect sequencing data of RNP transformed protoplast at 3 days post transformation.

PCR amplification and Sanger sequencing analysis of RNP transformed protoplasts at 3 days post transformation (DPT) for *PDS* editing showed 14.3% and 16.7% mutation rate for LbCas12aU/crRNA and ErCas12a/crRNA, respectively.

We first tested the mutation rate of the *LOB1* gene for LbCas12aU/crRNA and ErCas12a/crRNA RNPs. PCR amplification and Sanger sequencing analysis of RNP transformed protoplasts at 3 DPT for *LOB1* editing showed 48.1% and 34.8% mutation rate for LbCas12aU/crRNA and ErCas12a/crRNA, respectively. The mutation rate seems to associate with the quality and status of the embryogenic protoplasts because deep sequencing analysis of LbCas12aU/crRNA RNP transformed protoplasts from a different batch at 3 DPT demonstrated 71% mutation rate.

Related to the above point, fig. 3 summarizes the editing results for the citrus gene CsLOB1 that confers susceptibility to citrus canker disease. It is unclear how the authors selected the 42 regenerated plantlets, most of which carried biallelic mutations in CsLOB1. The biallelic mutation rate is strikingly high in this report, 97.4% for CsLOB1. Is this high biallelic mutation rate also observed with the CsPDS gene, if both green and albino embryos were counted? If such an observation holds true for both genes, it would suggest that citrus is quite different from other plants in response to the Cas12a/crRNA RNP system. Such a high frequency of obtaining biallelic alleles would make it an ideal plant model of both fundamental studies and practical applications.

Response: Actually, we were as surprised as you when we saw the sequencing result for the *LOB1* edited plants.

Please let me describe the process and how we got there.

Transgene-free editing of *LOB1* in citrus was initiated in my lab in 2017. We tested both plasmid and RNP methods for both Cas9/gRNA and Cas12a/crRNA, but did not get any genome edited citrus plants in the first 2 years. We then began to focus on Cas12a and include the *PDS* gene for editing owing to their obvious albino phenotype. We apologized that we did not calculate the mutation rate for *PDS* gene because we were really stressed out then that we have not made any biallelic/homozygous mutants in more than 2 years. We were also told by others that albino phenotype may randomly happen in plants regenerated from protoplasts. With that in mind, we have not included many green embryos in our sequencing analyses for embryos generated from *PDS* edited protoplasts, but only focused on sequencing the embryos showing albino phenotypes. We did sequencing analysis of RNP transformed protoplasts (3 days after RNP transformation) for *PDS* editing. We have added the following: PCR amplification and Sanger sequencing analysis of RNP transformed protoplasts at 3 days post transformation (DPT) for *PDS* editing showed 14.3% and 16.7% mutation rate for LbCas12aU/crRNA and ErCas12a/crRNA, respectively.

PCR and sequencing analysis of the *PDS* gene of the embryos actually showed that all our albino embryos (73 in total) contained mutations as guided by Cas12a/crRNA, instead of random mutations. Anyway, the *pds* experiment gave us strong hopes to continue even though it was already 2020 then, over three years since we started this project and even though we still did not get transgene-free *LOB1* edited citrus varieties.

We then focused our effort to edit *LOB1* using Cas12a/crRNA RNA. Sequencing analysis of RNP transformed protoplasts (3 days after RNP transformation) for *LOB1* editing a mutation rate of 48.1% for *LOB1*. Another culture of protoplasts showed a mutation rate of 70% for *LOB1*. Finally, in 2022, we

began to successfully generate plants from RNA transformed protoplasts for LOB1 editing and generated 39 lines (not including 3 died in the process) by the time we submitted the paper and several more afterward.

So here are our responses to your questions:

1. The 42 plantlets were all the regenerated lines that we have gotten. 3 plantlets did not survive after micrografting. We had 39 plantlets survived when we submitted the paper.
2. The biallelic mutation rate for CsPDS was much lower than LOB1 as shown in our sequencing data above.

Fig. 3: A: more details should be included for the panel, e.g. what do color and underline indicate? Related to this point, please check other figures and add more details in the figure legends when necessary.

Response: Revised as suggested.

Panel: B-F, images of three edited lines were shown yet the authors described in the legend that the data were similar from six lines. It would be great to show all six lines. The mutant line used to generate data for panels C-F should be indicated.

Response: We have included the three extra lines below.

Mutant line #5 was used for panels C-F.

For panel E, Zou et al (The Plant Journal (2021) 106, 1039–1057) showed that CsLOB1 RNAi plants had reduced Xcc growth in a *C. sinensis* cultivar. However, no difference was observed here between the control plant and the Cslob1 mutants with Xcc infection. Perhaps the authors should do a time course bacterial growth rather than just using one time point.

Response: We thank the reviewer for the excellent suggestion. During our trouble shooting discussion, we realized that in our original experiment for Xcc growth, Xcc at 10E8 CFU/ml was mistakenly used instead of the regularly used lower concentrations such as 10E7 CFU/ml. We have redone the Xcc titer assays using a time course experiment as suggested with the Xcc inoculum concentration at 10E7

CFU/ml. Xcc titer was significantly reduced at 9 days post inoculation, but only slightly reduced in earlier time points.

For Panel F: The authors mentioned that expression of some cell wall genes was up-regulated by CsLOB1 during Xcc infection, in the cslob1 mutant and wild-type *C. sinensis* cv. Hamlin. However, panel F seems to only show one condition. This reviewer suggests that both conditions, infection and non infection, should be tested for these gene expression.

Response: Thanks for the suggestion. We have done the suggested experiment as the new 4A.

Fig. 4b: Did the author check gene expression with and without Xcc infection (see comments for Fig. 3F)? Again, please indicate which transgenic line was used.

Response: We have decided not to include the gene expression data for PR genes because we need to conduct a more comprehensive study to understand the resistance mechanism of the lob1 mutant. Because the focus of the manuscript is generation of transgenic free LOB1 edited citrus plants. We think it is beyond the scope of this study to conduct a more comprehensive study to understand the resistance mechanism of the lob1 mutant.

Line 142: Please rephrase the sentence.

Response: Revised as follows: Next, we investigated whether our transgene-free lines contained off-target mutations.

Line 160-166: CsLOB1 was previously known to affect expression of cell wall related genes. Please tone down the importance of the work here – this work did not add to the understanding of canker resistance mechanism mediated by CsLOB1.

Response: We added the following: ...consistent with previous studies^{50,51,53}

Line 264: Two ErCas12a proteins were described. Please indicate clearly which one was used in the related experiments.

Response: We used ErCas12a and LbCas12a in our study. The description was clear on both proteins.

Line 299: Plant regeneration should be described in more details. Many abbreviations were used in Fig. S1, making it difficult to know the exact process of plant regeneration. Were there any chemicals used for selecting edited protoplasts and/or embryos?

Response: We have provided detailed information on the components of the media and the procedure.

Line 335-339: It is interesting that different Xcc concentrations and infection times were used for different gene expression analysis. Have the authors tested one Xcc concentration in a time course to see the progression of gene expression for genes of interest?

Response: We have removed the data related to PR genes, but redone the gene expression for cell wall related genes as suggested by the reviewer to include both wild type and lob1 mutants with and without Xcc inoculation at 9 days post inoculation.

Reviewers' Comments:

Reviewer #2:

Remarks to the Author:

I reviewed this manuscript previously. For most of my concerns, the authors have addressed the concern sufficiently and revised the manuscript accordingly. Based on my review and that of the other reviewers, there are still a few pieces of data that the authors do not seem to fully understand. However, the advance that is central to the manuscript appears to be solid (and a long time coming!).

Reviewer #4:

Remarks to the Author:

I did not see the paper before but I was asked as gene editing specialist by the editor to evaluate the response/revisions of the authors to the concerns of the original reviewers 1 and 3. Well, I also have to say that biallelic editing frequencies of over 90% with using RNPs is miraculous for plants. The authors were extremely lucky and I would not expect at all that they are able to obtain similar results in Citrus with other genes. As stated by them correctly the mutation itself might cause a selection advantage for plant transformation/regeneration. I also think that the authors should avoid the impression that the use of RNPs would - as a general rule - cause less off-site mutations. The nature of the endonuclease should be the same independent of its origin, it is the amount of the enzyme over time that counts. And here at least in many previous experiments there was a direct correlation between lower activity for on and off sites with DNA as well as with RNPs. In any case, this is an important contribution. It is great that the authors got cranker-resistant citrus plants by editing that are classified as non-GMOs. I am also convinced that they developed an efficient editing protocol for citrus but they should avoid the impression that such frequencies can be also obtained with other genes until they demonstrate the opposite themselves.

RESPONSE TO REVIEWERS' COMMENTS

Reviewer #2 (Remarks to the Author):

I reviewed this manuscript previously. For most of my concerns, the authors have addressed the concern sufficiently and revised the manuscript accordingly. Based on my review and that of the other reviewers, there are still a few pieces of data that the authors do not seem to fully understand. However, the advance that is central to the manuscript appears to be solid (and a long time coming!).

Response: We thank the reviewer for the positive comments.

Reviewer #4 (Remarks to the Author):

I did not see the paper before but I was asked as gene editing specialist by the editor to evaluate the response/revisions of the authors to the concerns of the original reviewers 1 and 3.

Well, I also have to say that biallelic editing frequencies of over 90% with using RNPs is miraculous for plants. The authors were extremely lucky and I would not expect at all that they are able to obtain similar results in Citrus with other genes. As stated by them correctly the mutation itself might cause a selection advantage for plant transformation/regeneration. I also think that the authors should avoid the impression that the use of RNPs would - as a general rule - cause less off-site mutations. The nature of the endonuclease should be the same independent of its origin, it is the amount of the enzyme over time that counts. And here at least in many previous experiments there was a direct correlation between lower activity for on and off sites with DNA as well as with RNPs.

In any case, this is an important contribution. It is great that the authors got cranker-resistant citrus plants by editing that are classified as non-GMOs. I am also convinced that they developed an efficient editing protocol for citrus but they should avoid the impression that such frequencies can be also obtained with other genes until they demonstrate the opposite themselves.

Response: We thank the reviewer for the insightful comments.

To address the reviewer's comments, we have added "It remains to be determined whether such a high editing efficacy can be achieved for other citrus genes beyond *LOB1*."